# SEMPANet: A Modified Path Aggregation Network with Squeeze-Excitation for Scene Text Detection

**DOI:** 10.3390/s21082657

**Published:** 2021-04-09

**Authors:** Shuangshuang Li, Wenming Cao

**Affiliations:** Guangdong Key Laboratory of Intelligent Information Processing and Shenzhen Key Laboratory of Media Security, Shenzhen 518060, China; lishuangshuang2016@email.szu.edu.cn

**Keywords:** text detection, natural scene, feature fusion

## Abstract

Recently, various object detection frameworks have been applied to text detection tasks and have achieved good performance in the final detection. With the further expansion of text detection application scenarios, the research value of text detection topics has gradually increased. Text detection in natural scenes is more challenging for horizontal text based on a quadrilateral detection box and for curved text of any shape. Most networks have a good effect on the balancing of target samples in text detection, but it is challenging to deal with small targets and solve extremely unbalanced data. We continued to use PSENet to deal with such problems in this work. On the other hand, we studied the problem that most of the existing scene text detection methods use ResNet and FPN as the backbone of feature extraction, and improved the ResNet and FPN network parts of PSENet to make it more conducive to the combination of feature extraction in the early stage. A SEMPANet framework without an anchor and in one stage is proposed to implement a lightweight model, which is embodied in the training time of about 24 h. Finally, we selected the two most representative datasets for oriented text and curved text to conduct experiments. On ICDAR2015, the improved network’s latest results further verify its effectiveness; it reached 1.01% in F-measure compared with PSENet-1s. On CTW1500, the improved network performed better than the original network on average.

## 1. Introduction

The rapid development of deep learning has promoted the remarkable success of various visual tasks. Among them, the progress of text detection in natural scenes is increasing. Traditional CNN networks can effectively extract image features and train text classifiers. Other networks are gradually being derived from CNNs, such as segmentation, regression, and end-to-end methods. Deep learning brings algorithms that include more diverse structures, and the results are even more impressive [1,2].

Text detection in natural scenes is based on target detection, but it is different from target detection: it considers the diversity of text direction rotation and size ratio changes; the lighting of the scene, such as the actual streets and shopping mall scenes, (causing the image to be blurred); the inclined shooting angle; and the difficulty caused by the change of text language from horizontal text to curved text. The competition is still fierce. The disadvantage of most network structures is that the simple form cannot satisfy the improvement of the results. Generally speaking, models with high results have significant parameters and large models, while complex systems are time-consuming. Many algorithms are in the research stage, and it is difficult to enter the batch use stage, which still has a large unmet demand. Therefore, this type of application-based algorithm needs to produce state-of-the-art accuracy in theoretical research and consider the request for production in the application scenario and the lightweight model in the portable device.

A series of target detection algorithms [3,4] have been applied in the scene text detection field and promoted the research and development of natural scene text detection recently. The SSD algorithm [5] proposed by Liu et al. uses a pyramid structure and feature maps of different sizes to perform softmax classification and position regression on multiple feature maps simultaneously. The location box of the real target is obtained through classification and bounding box regression. Based on SSD, many researchers improve their methods for the detection of scene text. Shi et al. proposed the SegLink algorithm [6], which is enhanced based on the SSD target detection method. It detects partial fragments at first, and connects all fragments through rules to obtain the final text line, which can better detect text lines of any length. Ren et al. [7] proposed the Faster-RCNN target detection algorithm. Reference [2] proposed a hybrid framework that integrates Persian dependency-based rules and DNN models, including long short-term memory (LSTM) and convolutional neural networks (CNN). Tian et al. proposed the CTPN algorithm [8], which combines CNN and LSTM networks, and adds a two-way LSTM to learn the text-based sequence features via Faster-RCNN; this kind of approach is conducive to the prediction of text boxes. Ma et al. proposed the RRPN algorithm [9] based on Faster-RCNN, a rotation area suggestion network using text inclination angle information, which adjusts the angle information for border regression to fit the text area better.

It is worth noting that many new tasks based on ResNet [10,11] and FPN [12] have appeared and have attracted more attention in recent years. At the same time, ResNet and FPN have many improved methods. SENet [13] adds an SE module to the residual learning unit and integrates a learning mechanism to explicitly model the interdependence between channels so that the network can automatically obtain the importance of each feature channel. This importance enhances the valuable features and suppresses the features that are not useful for the current task. The SE module is also added to some target detection algorithms. Take M2Det [14] as an example: the SFAM structure in this paper uses an SE block to perform an attention operation on the channel to capture useful features better. PANet [15] uses the element addition operation by layer, different levels of information are fused, and a shortcut path is introduced. The bottom-up way is enhanced, making the low-level information more easily spread to the top, and the top-level can also obtain fine-grained local information. Each level is a richer feature map. It can be seen from the above that the latest improved methods also have apparent effects on the improvement of other tasks. Based on the above, this paper introduces a new basic network framework for scene text detection tasks, namely, SEMPANet.

Compared with the previous scene text detection systems, the proposed architecture has two different characteristics:

(1) Compared with the standard ResNet residual structure, the addition of SENet in this paper enables the network to enhance the beneficial feature channel selectively and suppress the useless feature channel by using the global information to realize the feature channel adaptive calibration, reflected in the improvement of the value in the experimental results.

(2) Considering the information flow between the network layers during the training period, the bottom-up path of MPANet is enhanced, making the bottom-up information more easily spread to the top. This paper verifies the influence of PANet on the detection method and modifies the process of PANet to make it more effective. Experimental results show that it can get a more accurate text detection effect than the model with FPN.

The paper is organized as follows:

Section 2 introduces the popular experimental framework in scene text detection in recent years, which describes related work from the following three aspects: whether the detector is based on anchoring, whether it is one stage or two-stage, and whether it is based on RESNET and FPN. Section 3 presents the overall network framework of this paper; the principle of the algorithm is introduced as well, including the SE module and MPANet module. Section 4 includes testing results and their evaluation by the proposed methods. Conclusions are given in Section 5.

## 2. Related Work

### 2.1. Anchor-Based and Anchor-Free

Both anchor-based detectors and anchor-free detectors have been used in recent natural scene text detection tasks.

Specifically, anchor-based methods traverse the feature maps calculated by convolutional layers, and place a large number of pre-defined anchors on each picture, the categories are predicted, and the coordinates of these anchors are optimized, which will be regarded as detection results. According to the text area’s aspect ratio characteristics, TextBoxes [16] equips each point with six anchors with different aspect ratios as the initial text detection box. TextBoxes+ [17] can detect text in any direction, which uses text boxes with oblique angles to detect irregularly shaped text. DMPNet [18] retains the traditional horizontal sliding window and separately sets six candidate text boxes with different inclination angles according to the inherent shape characteristics of the text: add two 45-degree rectangular windows in the square window; add two long parallelogram windows in the long rectangular window; add two tall parallelogram windows inside the tall rectangular window. The four vertices coordinates of the quadrilateral are used to represent the text candidate frame.

Anchor-free detectors can find objects directly in two different ways without defining anchors in advance. One method is to locate several pre-defined or self-learning key points and limit the spatial scope of the target. Another method is to use the center point or area of the object to define the positive, and then predict the four distances from the positive to the object boundary. For example, in FCOS [19], the introduction of centerness can well inhibit these low-quality boxes’ production. Simultaneously, it avoids the complex calculation of anchor frames, such as calculating the overlap in the training process, and saves the memory consumption in the training process. AF-RPN [20] solves the problem that the classic RPN algorithm cannot effectively predict text boxes in any direction. Instead of detecting fusion features from different levels, it detects the text size by the size of the multi-scale components extracted by the feature pyramid network. The RPN stage abandons the use of anchors and uses a point directly to return the coordinates of the four corners of the bounding box, and then shrinks the text area to generate the text core area.

PSENet [21] is slightly different from anchor-free methods. It segments the fusion features of different scales’ outputs by the FPN network. Each text instance is reduced to multiple text segmentation maps of different scales through the shrinkage method. The segmentation maps of different scales are merged by the progressive expansion algorithm based on breadth-first-search, which focuses on reconstructing the text instance as a whole to get the final detected text. The progressive scale expansion algorithm can detect the scene text more accurately and distinguish the text that is close or stuck together, which is another method that can process text well without an anchor.

### 2.2. One-Stage and Two-Stage Algorithms

The representative one-stage and two-stage algorithms are YOLO and Faster-R-CNN, respectively.

The most significant advantage of the single-stage detection algorithm is that it is fast. It provides category and location information directly through the backbone network without using the RPN network to display the candidate area. The accuracy of this algorithm is slightly lower than that of the two-stage. With the development of target detection algorithms, the accuracy of single-stage target detection algorithms has also been improved. Gupta et al. proposed the FCRN model [22], which extracts features based on the full convolutional network, and then performs regression prediction on the feature map by convolution operation. Unlike the prediction of a category label in FCN [23], it predicts the bounding box parameters of each enclosing word, including the center coordinate offset, width, height, and angle information. EAST [24] directly indicates arbitrary quadrilateral text based on the full convolutional network (FCN). It uses NMS to process overlapping bounding boxes and generates multi-channel pixel-level text scoring maps and geometric figures with an end-to-end model. R-YOLO [25] proposed a real-time detector including a fourth-scale detection branch based on YOLOv4 [26], which improved the detection ability of small-scale text effectively.

The precision of the two-stage is higher, while the speed is slower than that of the one-stage. The two-stage network extracts deep features through a convolutional neural network, and then divides the detection into two stages: The first step is to generate candidate regions that may contain objects through the RPN network, and complete the classification of the regions to make a preliminary prediction of the position of the target; the second step is to further accurately classify and calibrate the candidate regions to obtain the final detection result. The entire network structure of RRPN [9] is the same as Faster-R-CNN, which is divided into two parts: one is used to predict the category, and the other one is used to regress the rotated rectangular box to detect text in any direction. Its two-stage is embodied in the use of RRPN to generate a candidate area with rotation angle information, and then adding an RROI pooling layer to generate a fixed-length feature vector, followed by two layers fully connected for the classification of the candidate area. Mask TextSpotter [27] is also a two-stage text detection network based on Mask R-CNN [28], it replaces the RoI pooling layer of Faster-R-CNN with the RoIAlign layer, and adds an FCN branch that predicts the segmentation mask. TextFuseNet [29] merged the ideas of masktextspotter and Mask R-CNN to extract multi-level features from different paths to obtain richer features.

### 2.3. ResNet and FPN

In addition to the design and improvement of various target detection algorithms that focus on different positions, a detector that can be applied currently in either one stage or two stages usually has the following two parts: the backbone network and the neck part.

It comprises a series of convolution layers, nonlinear layers, and downsampling layers for CNN. The features of images are captured from the global receptive field to describe the images. VGGNet [30] improves performance by continuously deepening the network structure. The increase in the number of network layers will not bring about an explosion in the number of parameters, and the ability to learn features is more vital. The BN layer in batch normalization [31] suppresses the problem that small changes in parameters are amplified as the characteristic network deepens and is more adaptable to parameter changes. Its superior performance makes it the standard configuration in current convolutional networks. ResNet establishes a direct correlation channel between input and output. The robust parameterized layer concentrates on learning the residual between input and output, and improves gradient explosion and gradient disappearance when the network develops deeper.

The backbone of target detection includes VGG, ResNet, etc. In CTPN [8], the VGG16 backbone is first used for feature extraction, SSD network [5] also uses VGG-16 as the primary network. ResNet-50 module was first used for feature extraction in the method proposed by Yang et al. [32], and most of the later networks adopt the ResNet series. The backbone part has also helped develop many excellent networks, such as DenseNet. DenseNet establishes the connection relationship between different layers through feature reuse and bypass settings to further reduce the problem of gradient disappearance and achieve a good training effect, instead of deepening the number of network layers in ResNet and widening network structure in Inception to improve network performance. Besides, the use of the bottleneck layer and translation layer makes the network narrower and reduces the parameters, suppressing overfitting effectively. Some detectors use DenseNet as a backbone for feature extraction.

With the popularity of multi-scale prediction methods such as FPN, many lightweight modules integrating different feature pyramids have been proposed. In FPN, the information from the adjacent layers of bottom-up and top-down data streams will be combined. The target texts of different sizes use the feature map at different levels and detect them separately, leading to repeated prediction results. It is not possible to use the information of the other level feature maps. The neck part of the network has also further developed PANet and other networks. In the target detection algorithm, Yolov4 [26] also uses the PANet method based on the FPN module of YOLOv3 [33] to gather parameters for the training phase to improve the performance of its detector, which proves the effectiveness of PANet. That multi-level fusion architecture has been widely used recently.

## 3. Principle of the Method

This paper is based on PSENet: without an anchor and in one stage, it explores common text detection frameworks such as ResNet and FPN in other directions. The proposed framework is mainly divided into two modules: the SENet module and the MPANet module. In the residual structure of ResNet, the original PANet processes adjacent layers through addition operations. The MPANet used in this paper is modified from original PANet and connects the characteristic graphs of adjacent layers together to improve the effect. Figure 1 clearly describes the proposed architecture of the scene text detection algorithm.

### 3.1. SENet Block

Convolution neural networks can only learn the dependence of local space according to the receptive field’s size. A weight is introduced in the feature map layer considering the relationship between feature channels. In this way, different weights are added to each channel’s features to improve the learning ability of features. It should be noted that the SE module adds weights in the dimension of channels. YOLOv4 uses the SE module to do target detection tasks, proving that the SE module can improve the network.

In terms of function, the framework shown in Figure 2 consists of three parts: firstly, a backbone network is constructed to generate the shared feature map, and then a squeeze and excitation network is inserted. This framework’s key is adding three operations to the residual structure: squeeze feature compression, exception incentive, and weight recalibration.

Main steps of SENet:

(1) The spatial dimensions of features are compressed, and global average pooling is used Capture the global context, compress all the spatial information to generate channel statistics, compress the size of the graph from H × W to 1 × 1, and the one-dimensional parameter 1 × 1 can obtain the global view of H × W, and the perception area is wider, that is, the statistical information z, z ∈ R ${}^{C}$. The c-th element of z in the formula is calculated by the following formula:(1)$$z_{c}=F_{sq}\left(u_{c}\right)=\frac{1}{H\times W}\sum_{i=1}^{H}\sum_{j=1}^{W}u_{c}\left(i,j\right)$$
where F${}_{sq}$ (·) is the compression operation, and u${}_{c}$ is the c-th feature.

(2) A 1 × 1 convolution and Relu operation follow, reducing the dimension by 16 times from 256; that is, the channel is transformed to 16—Relu activation function δ(x) = max(0,x), dimension reduction layer parameter, $W_{1}\in R^{C\times \frac{C}{r}}$; then, the dimension increment layer of 1 × 1 convolution stimulates the number of channels to the original number of 256.
(2)$$S=F_{ex}\left(Z,W\right)=\sigma \left(g\left(Z,W\right)\right)=\sigma \left(W_{2}\delta \left(W_{1}Z\right)\right)$$
where the sigmoid activation function $\sigma \left(x\right)=\frac{1}{\left(1+e^{-x}\right)}$, and the dimension increase layer parameter $W_{2}\in R^{C\times \frac{C}{r}}$,$F_{ex}\left(\cdot \right)$ is the excitation operation, S = [$s_{1},s_{2},s_{3},...,s_{c}$], $s_{k}\in R^{H\times W}\left(k=1,2,3,...,c\right)$;

(3) The weight is generated for each feature channel’s importance after feature selection is obtained, which are multiplied one by one with the previous features to complete the calibration of the original features in the channel dimension.
(3)$${\tilde{X}}_{C}=F_{scale}\left(u_{C},s_{C}\right)=s_{C}\cdot u_{C}$$
where $\tilde{X}=\left[{\tilde{x}}_{1},{\tilde{x}}_{2},...,{\tilde{x}}_{C}\right]$, $F_{scale}\left(u_{C},s_{C}\right)$ refers to the corresponding channel product between the feature map $u_{C}\in R^{H\times W}$ and the scalar s${}_{C}$.

### 3.2. Architecture of MPANet

Inspired by FPN, which obtains the semantic features of multi-scale targets, we propose a path aggregation network described in Figure 3; it can be added to the FPN to make the features of different scales more in-depth and more expressive. The emphasis is on fusing low-level elements and adaptive features at the top level.

Our framework improves the bottom-up path expansion. We follow FPN to define the layer that generates the feature map. The same space size is in the same network stage. Each functional level corresponds to a specific stage. We also need ResNet-50 as the basic structure; the output vector of Conv2-x, Conv3-x, Conv4-x, and Conv5-x in the ResNet network is C${}_{2}$,C${}_{3}$,C${}_{4}$,C${}_{5}$. P${}_{5}$, P${}_{4}$, P${}_{3}$, and P${}_{2}$ are used to represent the feature levels from top to bottom of FPN generation.
(4)$$\begin{array}{c} P_{i}=\left\{\begin{array}{cc} f_{1}^{3\times 3}\left(C_{i}\right) & i=5.\\ f_{2}^{3\times 3}\left\{C_{i}⊕F_{upsample}^{\times 2}\left[f_{1}^{3\times 3}\left(P_{i+1}\right)\right]\right\} & i=2,3,4. \end{array}\right. \end{array}$$
where $f_{1}^{3\times 3}$ means that each P${}_{i+1}$ first passes a 3 × 3 convolutional layer to reduce the number of channels; then the feature map is upsampled to the same size as C${}_{i}$ and adds to the C${}_{i}$ feature map elements; $f_{2}^{3\times 3}$ means that the summed feature map undergoes another 3 × 3 convolution operation to generate P${}_{i}$.
(5)$$\begin{array}{c} N_{i}=\left\{\begin{array}{cc} P_{i} & i=2.\\ f_{2}^{3\times 3}\left\{f^{1\times 1}\left[P_{i+1}∥f_{1}^{3\times 3}\left(N_{i}\right)\right]\right\} & i=3,4,5. \end{array}\right. \end{array}$$

Our augmented path starts from the bottom P${}_{2}$ and gradually approaches P${}_{5}$. The spatial size is gradually sampled down by factor 2 from P${}_{2}$ to P${}_{5}$. We use N${}_{2}$,N${}_{3}$,N${}_{4}$,N${}_{5}$ to represent the newly generated feature graph. Note that N${}_{2}$ is P${}_{2}$, without any processing, and retains the original feature map’s information.

As shown in Figure 3, each building block needs a higher resolution feature map N${}_{i}$ and a coarser P${}_{i+1}$ to generate a new feature map N${}_{i+1}$.

$f_{1}^{3\times 3}$ means that each feature map N${}_{i}$ passes through a 3×3 convolution layer with a step size of 2 to reduce the space size firstly.

“‖“ means that the feature map P${}_{i+1}$ of each layer is connected horizontally, not added, but concatenated with the downsampled map.

After this operation, $f^{1\times 1}$ means that the number of channels in the concatenated feature map will be doubled, through 1×1 convolution layer, the step size is 1, and then the channel number is restored to 256.

$f_{2}^{3\times 3}$ means that the fused feature map is then processed by 3×3 convolution fusion to generate N${}_{i+1}$ layer for the next step. This is an iterative process, which ends when it approaches P${}_{5}$. In these building blocks, we mostly use each feature map with 256 channels.
(6)$$N=N_{2}∥F_{upsample}^{\times 2}\left(N_{3}\right)∥F_{upsample}^{\times 4}\left(N_{4}\right)∥F_{upsample}^{\times 8}\left(N_{5}\right)$$

Then, the suggestions of each function are collected from the new feature mapping, namely, N${}_{2}$,N${}_{3}$,N${}_{4}$,N${}_{5}$. The N${}_{3}$, N${}_{4}$ and N${}_{5}$ are upsampled to the size of N${}_{2}$, $F_{upsample}^{\times 2}$ , $F_{upsample}^{\times 4}$, $F_{upsample}^{\times 8}$ refers to 2, 4, 8 times unsampling, and the four layers are concatenated into a feature map.
(7)$$input_{PSE}=F_{upsample}^{\times 2}\left\{f^{1\times 1}\left[f^{3\times 3}\left(N\right)\right]\right\}$$
where $f^{3\times 3}$ refers to convolution operation for reducing the number of channels to 256, $f^{1\times 1}$ refers to the generation of 7 segmentation results. F${}_{upsample}$ refers to upsampling to the size of the original image, and the output channel is 7, which is input into the PSE block.

## 4. Experiments

### 4.1. Experiment Configuration

The computer configuration shows in Table 1, the training details are as follows:

When training ICDAR2015 [34] and CTW1500 [35] datasets separately, we use a single dataset, note that there are no extra data available for pretraining, e.g., SynthText [22] and IC17-MLT [36]. Before loading them into the network for training, we preprocess images with data augmentation, images are rescaled and returned with random ratios of 0.5,1.0,2.0,3.0; the rotated images randomize in the range [$-10^{\circ },10^{\circ }$]. Samples are randomly selected from the transformed images, and the minimum output area of the bounding box is calculated for ICDAR2015, the final result is generated by PSE results for CTW1500. All the networks are using SGD. We train each independent dataset with a batch size of 10 on two GPUs for 600 iterations. The training time for each lightweight model is only 24 h. The initial learning rate is set to $1\times 10^{-3}$, divided by 10 at 200 and 400 iterations. We ignore all the text areas labeled as “DO NOT CARE“ in the dataset during the training stage, which are not shown as data. Other hyper-parameter settings of the loss function are consistent with PSENet, such as the number of λ is set to 0.7, the positive value of ohem is set to 3, etc. During the testing stage, the confidence threshold is set to 0.89.

### 4.2. Benchmark Datasets

#### 4.2.1. ICDAR2015

This is a standard dataset proposed for scene text detection in the Challenge4 of ICDAR2015 Robust Reading Competition, which is divided into two categories: the training part contains 1000 image-text pairs; the testing part contains 500 image-text pairs. Each picture is associated with one or more labels annotated with four vertices of the quadrangle. Unlike the previous datasets (such as ICDAR2013 [37]) that only contain horizontal text, the orientations of the reference text in this benchmark are arbitrary.

#### 4.2.2. CTW1500

It is a challenging text detection dataset in long curve format, 1000 for training and 500 for testing form a total of 1500 images. Unlike traditional text datasets (such as ICDAR2017 MLT), the text instance in CTW1500 is marked by a 14-point polygon. The annotations in this dataset are labelled in textline level, which can describe the arbitrary curved form.

### 4.3. Performance Evaluation Criteria

In this detection algorithm, three evaluation indexes are involved, namely:

#### 4.3.1. Recall

Recall rate(R) is the ratio of the number of positive classes predicted as positive classes to the number of positive real positive classes in the dataset, that is, how much of all the accurate text has been detected.
(8)$$recall=\frac{TP}{TP+FN}$$

#### 4.3.2. Precision

The precision rate(P) represents the ratio of all samples to the total number of samples predicted correctly, that is, how much text detected is accurate.
(9)$$precision=\frac{TP}{TP+FP}$$

#### 4.3.3. F-measure

We aim to have higher precision and recall in the evaluation results, but they are rarely in high results at the same time. Generally speaking, the former is higher while the latter is often inclined to the lower side; the latter is higher while the former is usually lower.

Therefore, when considering the performance of the algorithm, the precision rate and recall rate are not unique. We need to link the two to evaluate. Generally, the weighted average of the two is used to measure the quality of the algorithm and reflect the overall index, namely, F-measure(F). The formula is as follows:(10)$$\frac{2}{F}=\frac{1}{precision}+\frac{1}{recall}$$
the formula is transformed to:(11)$$F=\frac{2PR}{P+R}=\frac{2TP}{2TP+FP+FN}$$

Here, TP, FP, and FN are the numbers of True Postive(the instance is a positive class while the prediction is a positive class), False Postive(the instance is a negative class while the prediction is a positive class), and False Negative(the instance is a positive class while the prediction is a negative class), respectively.

### 4.4. Ablation Study

#### 4.4.1. Effects of MPANet

We conduct several ablation studies on ICDAR2015 and CTW1500 datasets to verify the effectiveness of the proposed MPANet(see Table 2). Note that all the models are trained using only official training images. As shown in Table 2, MPANet obtains 1.01% and 1.21% improvement in F-measure on ICDAR2015 and CTW1500, respectively.

Figure 4 shows the train loss difference between modified PANet with SE block (SEMPANet) and MPANet without SE block (MPANet). It demonstrates that the loss function of SEMPANet drops faster on ICDAR2015. Figure 5 shows the loss comparison of two models with and without SE block, which proves that the loss function of MPANet model has a slightly faster convergence effect on average than the other one on CTW1500. The difference of the loss function on the two datasets is reflected in the last two rows of Table 4 and Table 5.

#### 4.4.2. Effects of the Threshold λ in the Testing Phase

The hyper-parameter λ in the final test balances the influence between the three evaluation indexes. Table 3 compares the prediction effects of MPANet and SEMPANet with different λ within a short fluctuation range on the dataset ICDAR2015. We see that when SEMPANet with a λ of 0.89 is used, even if the performance is robust to changes in λ, in the average performance of the three evaluation indexes, F-measure is higher than PSENet, and Recall also performs best.

### 4.5. Experimental Results

#### 4.5.1. Evaluation on Oriented Text Benchmark

In order to verify the effectiveness of the bankbone proposed in this paper, we have carried out comparative experiments on ICDAR2015 with CTPN, Seglink, EAST, PSENet and other mainstream methods. The ICDAR2015 dataset mainly includes horizontal, vertical and slanted text. As shown in Table 4, the proposed method without external data achieves a state-of-the-art result of 80.45%, 82.80% and 81.61% in recall, precision and F-measure, respectively. Each paper in Table 4 has its representative detection method for natural scene text characteristics. Compared with EAST, our precision is reduced by 0.8%, while recall and F-measure are increased by 6.95% and 3.41%, respectively. Compared with WordSup, the recall, precision and F-measure are increased by 3.45%, 3.5% and 3.41%, respectively. Compared with PAN, our precision is slightly decreased by 0.1%, while recall is increased by 2.65%, the F-measure reflecting the comprehensive detection ability is increased by 1.31%. We have also compared with several lightweight networks in 2020. As shown in Table 3, we selected the results of three indicators that have been improved to above 80 when considering the overall performance. Compared with [38,39,40], our recall are increased by 3.75% 0.25% and 0.77%, respectively.

Compared with PSENet-1s, we can find that this paper’s method has improved recall, precision, and F-measure. The rates are increased by 0.75%, 1.3% and 1.01%, respectively. The comparison with the above methods on the ICDAR2015 dataset shows that the method proposed in this paper has a high level of detection results for regular text and slanted text. Overall, SEMPANet has a higher recall rate than MPANet on ICDAR2015, and its recall also achieves state-of-the-art result in Table 4. Some qualitative results are visualized in Figure 6.

#### 4.5.2. Evaluation on Curve Text Benchmark

We have verified the superiority of our method in the Curve text by conducting experiments on the public dataset CTW1500. The experimental results are shown in Table 5. The data for the comparison methods in the table are all from their corresponding papers. The CTW1500 dataset contains many curved letters. Methods such as CTPN and Seglink often fail to detect and label with rectangular boxes accurately. The bankbone proposed in this paper extracts richer features, combined with the post-processing part of PSENet, which is not limited by rectangular boxes and can detect any shape well. Compared with the benchmark method CTD+TLOC of the CTW1500 dataset, our accuracy rate has been improved by 3.02%, 6.68%, and 4.64% in recall, precision and F-measure, respectively. Compared with TextSnake, our recall is lower, while the precision is higher, which is 16.2% higher than TextSnake. The F-measure is lower by 2.16% compared with TextSnake. Compared with [38,40], our precision are increased by 2.28% and 3.48%, respectively.

Compared with PSENet-1s, the method proposed in this paper has a lower recall of 2.78%, however, the precision is greatly improved 3.48%. Due to the fact that many letters in the CTW1500 dataset are too close or even glued and overlapped, they are still difficult to separate. The F-measure of the method proposed in this paper reached 78.04%, indicating that it can detect curved text well. Figure 7 demonstrates some detection results of SEMPANet on CTW1500.

### 4.6. Discussion of Results

Most of the text can be well detected: see the green text detection boxes in Figure 6 and Figure 7. Invalid examples are shown in the red boxes in Figure 6b and Figure 7b, some of which are missing. We have analyzed the failure results of the proposed method. The following briefly introduces several sets of test results and analyzes environmental factors. In Figure 6b, the first image shows multiple text targets on the billboard. The red target of overly large size cannot be detected correctly, which is mistakenly divided into three target boxes. Due to the influences of the text and the background environment, the characters on the building in the second picture are omitted; due to the impact of the surrounding colors and the dense arrangement, the characters “3“ and “20“ in the third picture were left out. In Figure 7b, the “HO“ in the first image is omitted; the two small samples in the second image are omitted; characters in the third image are close to the white background. In short, the test results in an austere environment are good. For example, for text with a complex environment, a small portion of the text with shallow definition can be detected. Since there are scenes with many lines and colorful spots in the image, the existing model will classify the text as clearly recognizable by the human eye but not detected as background.

The proposed method can achieve outstanding detection results. However, PSENet still has limitations in processing small-sized text. Compared with the previous methods, this paper uses SEMPANet to improve the overall structure and adjusts the network parameters. In ICDAR2015, the recall rate R has been improved; P and F perform well; there are still deficiencies in the curved text CTW1500.

## 5. Conclusions

In this paper, our network can be divided into two parts: feature extraction and post-processing. The post-processing part using the progressive expansion algorithm can guarantee the accuracy of text detection, but the experimental results prove that the simple use of FPN network in the feature extraction part has insufficient feature extraction, which leads to the decline of the text detection effect. This paper proposes a new scene text detection method based on feature fusion. This method uses SENet as the basic network and integrates the features of the MPANet to make up for the lack of features extracted from the original network. The fusion strategy proposed in this paper enables the text detection model to reach a detection level higher than that of the original network. Finally, the progressive expansion algorithm is used for post-processing so that the entire model can detect the text quickly and accurately. With the aim of improving the experimental results, the method in this paper avoids the introduction of end-to-end networks with too many parameters, and finally achieves the purpose of accurate and fast text detection, which is of great significance for the research of natural scene text detection technology oriented toward actual application scenarios. Furthermore, I hope to introduce new mathematical tools for research and discussion. In that regard, a recent approach based on geometric algebra [48] extracts features for multispectral images to be investigated. Finally, other multi-dimensional data processing such as L1-norm minimization [49] and hashing networks [50] remain primarily unexplored and can benefit from further research.

## Figures and Tables

**Figure 1 sensors-21-02657-f001:**
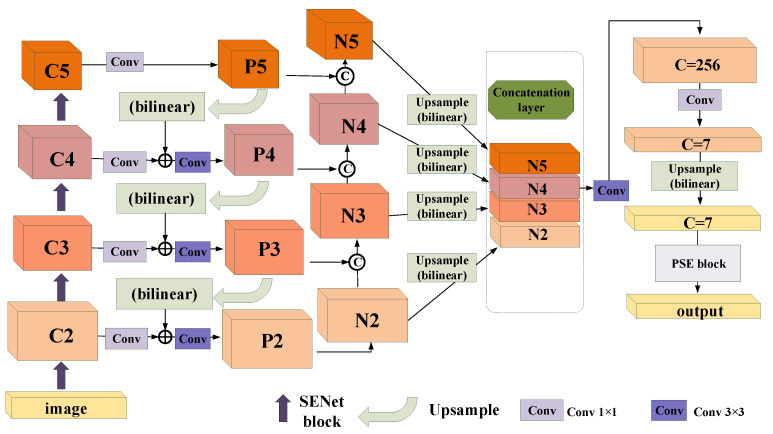
An illustration of our framework. It includes a basic structure with SE blocks; a backbone of feature pyramid networks; bottom-up path augmentation; the progressive scale expansion algorithm, which predicts text regions, kernels, and similarity vectors to describe the text instances. Note that we omit the channel dimensions of feature maps for brevity.

**Figure 2 sensors-21-02657-f002:**
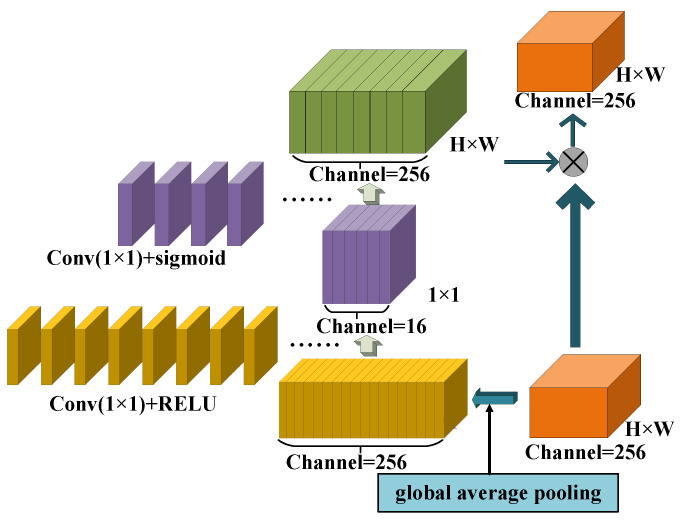
Illustration of an SE block in our model.

**Figure 3 sensors-21-02657-f003:**
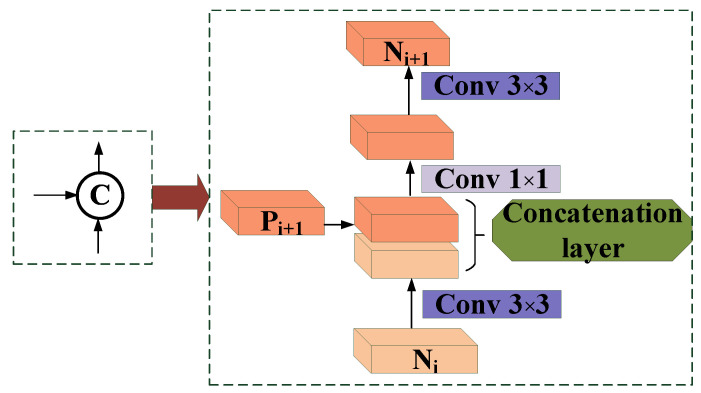
An illustration of our modification of the bottom-up path augmentation.

**Figure 4 sensors-21-02657-f004:**
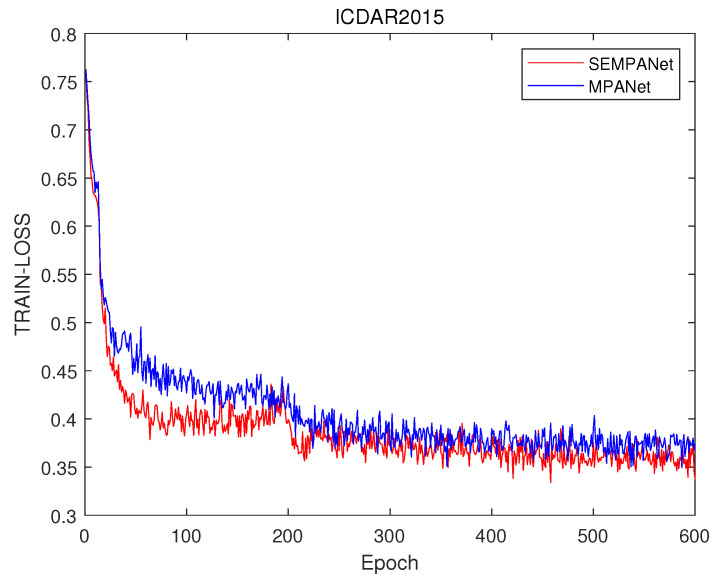
Ablation study of an SE block on ICDAR2015. These results are based on (ResNet 50 and SE block) and (ResNet 50 block) trained on MPANet.

**Figure 5 sensors-21-02657-f005:**
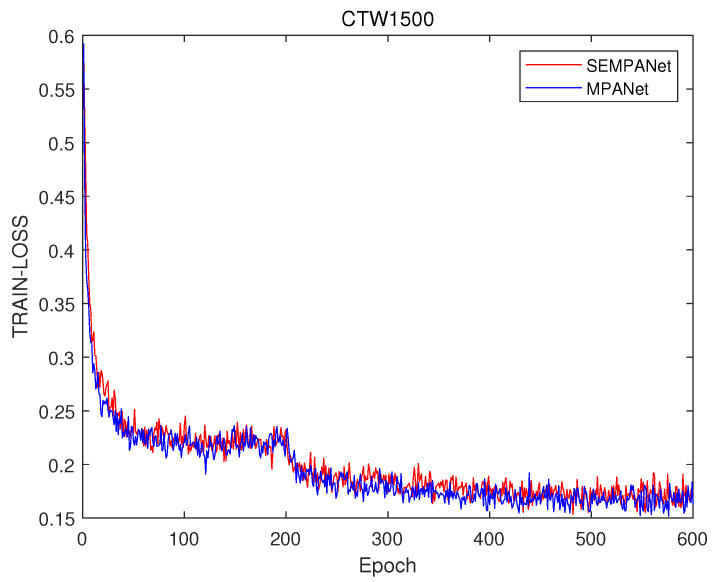
Ablation study of an SE block on CTW1500. These results are based on (ResNet 50 and SE block) and (ResNet 50 block) trained on MPANet.

**Figure 6 sensors-21-02657-f006:**
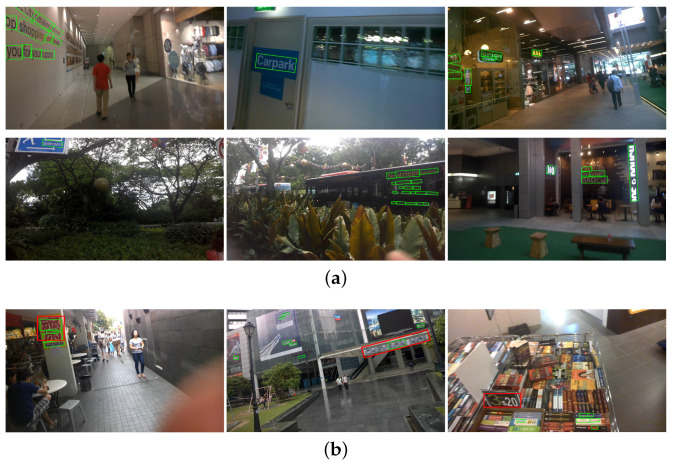
Results on ICDAR2015. The green boxes in (**a**,**b**) and the red boxes in (**b**) represent the evaluation results of the text and the error detection boxes of them, respectively.

**Figure 7 sensors-21-02657-f007:**
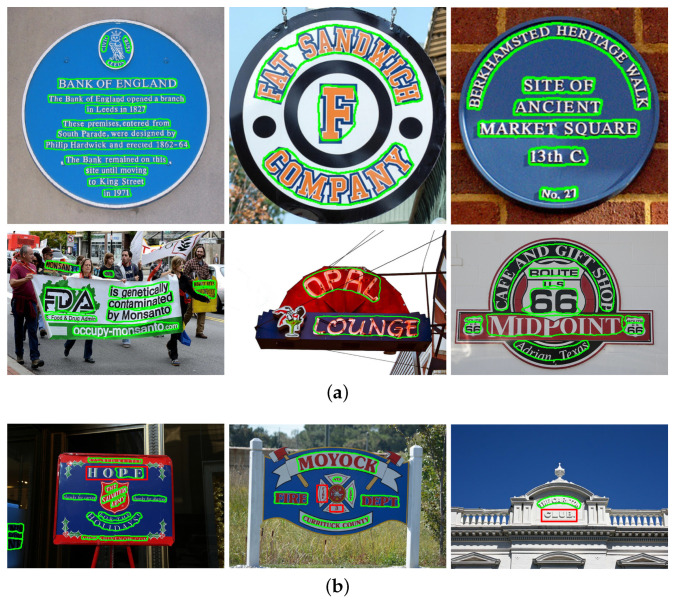
Some visualization results from CTW1500. The green boxes in (**a**,**b**) and the red boxes in (**a**) represent the evaluation results of the text and the error detection boxes of them respectively.

**Table 1 sensors-21-02657-t001:** Computer configuration.

Software Platform	System	Code Edit	Framework
	Ubuntu 16.04 LTS	Python2.7	PyTorch1.2
**Hardware Platform**	**Memory**	**GPU**	**CPU**
	25 GB	GeForce RTX 2080Ti 11G memory	28 core

**Table 2 sensors-21-02657-t002:** The performance gain of MPANet. * and † are results from ICDAR2015 and CTW1500, respectively. FPN * and FPN † represent the results of using the FPN network model in PSE [21] on ICDAR2015 and CTW1500, respectively.

Method	Recall	Precision	F-Measure
FPN *	79.68	81.49	80.57
MPANet *	79.97	83.26	81.58
Gain *	0.29	1.77	1.01
FPN †	75.55	80.57	78.00
MPANet †	75.52	83.29	79.21
Gain †	−0.03	2.72	1.21

**Table 3 sensors-21-02657-t003:** The performance comparison of λ.

λ in MPANet	Recall	Precision	F-Measure
0.93	78.77	85.92	82.19
0.91	79.82	84.25	81.98
0.89	79.97	83.26	81.58
**λ in SEMPANet**	**Recall**	**Precision**	**F-Measure**
0.93	78.57	84.74	81.54
0.91	79.83	83.57	81.65
0.89	80.45	82.80	81.61

**Table 4 sensors-21-02657-t004:** The single-scale results on ICDAR2015. “Ext“ indicates external data. MPANet is a model without an SE module.

Method	Year	Ext	Recall	Precision	F-Measure
CTPN [8]	2016	**-**	51.6	74.2	60.9
Seglink [6]	2017	*√*	73.1	76.8	75.0
SSTD [41]	2017	*√*	73.9	80.2	76.9
EAST [24]	2017	**-**	73.5	83.6	78.2
WordSup [42]	2017	*√*	77.0	79.3	78.2
DeepReg [43]	2017	**-**	80.0	82.0	81.0
RRPN [9]	2018	**-**	73.0	82.0	77.0
Lyu et al. [44]	2018	*√*	70.7	94.1	80.7
PAN [45]	2019	**-**	77.8	82.9	80.3
PSENet-1s [21]	2019	**-**	79.7	81.5	80.6
Pelee-Text++ [39]	2020	*√*	76.7	87.5	81.7
Qin et al. [40]	2020	-	80.20	82.86	81.56
Jiang et al. [38]	2020	-	79.68	85.79	82.62
MPANet		**-**	79.97	83.26	81.58
SEMPANet		-	80.45	82.80	81.61

**Table 5 sensors-21-02657-t005:** The single-scale results from CTW1500. * indicates the results from [35]. Ext is short for external data used in the training stage. MPANet is a model without an SE module.

Method	Year	Ext	Recall	Precision	F-Measure
CTPN * [8]	2016	-	53.8	60.4	56.9
Seglink * [6]	2017	-	40.0	42.3	40.8
EAST * [24]	2017	-	49.1	78.7	60.4
CTD+TLOC [35]	2017	-	69.8	77.4	73.4
TextSnake [46]	2018	*√*	85.3	67.9	75.6
CSE [47]	2019	*√*	76.0	81.1	78.4
PSENet-1s [21]	2019	-	75.6	80.6	78.0
Jiang et al. [38]	2020	-	75.9	80.6	78.2
Qin et al. [40]	2020	-	76.8	81.8	79.4
MPANet		-	75.52	83.29	79.21
SEMPANet		-	72.82	84.08	78.04

## Data Availability

Not applicable.

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
