# Peer review of "SEMPANet: A Modified Path Aggregation Network with Squeeze-Excitation for Scene Text Detection"

_sensors, 2021, doi:10.3390/s21082657_

Round 1
Reviewer 1 Report
Abstract is little bit fuzzy and it does not put the strong impression on the reader. Be more precise and concise regarding contribution.
Introduction needs heavy polishing. It simply does not have proper flow of information toward contribution part.
Related work is missing some important references. I will just mention two papers [1-2].
Result section has a number of issues.
1) Obtained results are not compared with the state-of-the art. The obtained results (F1 measure) is below current results obtained by the most recent methods.
2) Limited results (only on two databases)
3) Discussion is weak.
4) FPS is not reported.
Generally speaking, the proposed method does not have better F1 measure nor it has higher processing speed in comparison with the state of the art.
Language
Generally speaking the language and writing style is at low level. I am not English native speaker, however on number of sentences I am struggling to get the meaning. Therefore it is very difficult to judge about actual paper contribution.
Unclear or not enough concise language:
Line 5-6 good effect?
Line 11 relative improvement from what basis?
Line 21-22
Line 26
Line 28 - 31
Line 56
...
The rest of the paper follows the same pattern of writing.
[1] R-YOLO: A Real-Time Text Detector for Natural Scenes with Arbitrary Rotation
[2] TextFuseNet: Scene Text Detection with Richer Fused Features
Author Response
We sincerely thank you for your suggestions. Through the suggestions, I have made changes to the relevant parts of the paper, and organized all the replies corresponding to the suggestions into a word document.

Reviewer 2 Report
Minors:
Fig.1, Fig.3 - please reformat images - they are ugly and chaotic. Background should be removed.
Fig.2 Background should be removed. \times operator should be used instead *
sometimes h x w, sometimes W x H
Main Steps <- Main steps
YOLOV4 <- YOLOv4
and more... - please clarify notation in this paper
"4. EXPERIMENTAL AND RESULTS" <- please fix language mistake
Table 1
no CPU data
Memory 25G (bits/bytes ?)
Code Edit <- ???
Table 2
78 <- 78.00
with & without <- with and without
" “P”, “R” and “F” represent the precision, recall and F-measure respectively. " <- it should be inside this table directly (1'st row)
Majors:
1.
"Figure 7 demonstrates some detection results of SEMPANet on CTW1500."
more comparative results should shown - not only good detections.
2.
Table 2 shows improvement over rest methods.
The main question is about wrong detections for particular method:
- Are the valid detections a subset of the valid detections of the proposed method or not?
-Are two other methods together better than the proposed method?
Author Response

(The authors gave the same response as above.)

Reviewer 3 Report
This paper study the problem of scene text detection, improve it on the basis of ResNet and FPN, and propose a SEMPANet framework which is more conducive to the combination of feature extraction in the early stage
The novelty of the paper is not clear
I would recommend to compare the results with other methods such as CNN, LSTM, BiLSTM and 1D-CNN.
There is lack of comparision with state-of-the-art approaches such as:
Aljuaid, H., Iftikhar, R., Ahmad, S., Asif, M. and Afzal, M.T., 2021. Important citation identification using sentiment analysis of In-text citations. Telematics and Informatics, 56, p.101492.
Dashtipour, K., Gogate, M., Li, J., Jiang, F., Kong, B. and Hussain, A., 2020. A hybrid Persian sentiment analysis framework: Integrating dependency grammar based rules and deep neural networks. Neurocomputing, 380, pp.1-10.
In addition, there is lack of experimental setup, and also the confusion matrix and training time for the model should be added
Author Response

(The authors gave the same response as above.)

Round 2
Reviewer 1 Report
Authors tried to address issues that I raised in previous review.
However, I still think that
- the method should be compared with the state of the art methods (whether we are talking about F1 and/or FPS)
- language and style should be improved
- related work and how actually proposed method supplement state-of-the art is still not completely described
Reviewer 2 Report
ok
Author Response
Thanks for your advice. We reviewed the full text again and revised some sentences.